# Comparison of Mitochondrial Genomes between a Cytoplasmic Male-Sterile Line and Its Restorer Line for Identifying Candidate CMS Genes in *Gossypium hirsutum*

**DOI:** 10.3390/ijms23169198

**Published:** 2022-08-16

**Authors:** Lisha Xuan, Guoan Qi, Xiaoran Li, Sunyi Yan, Yiwen Cao, Chujun Huang, Lu He, Tianzhen Zhang, Haihong Shang, Yan Hu

**Affiliations:** 1Institute of Crop Science, College of Agriculture and Biotechnology, Zhejiang University, Hangzhou 310058, China; 2Hainan Institute of Zhejiang University, Yazhou Bay Science and Technology City, Sanya 572025, China; 3Zhengzhou Research Base, State Key Laboratory of Cotton Biology, School of Agricultural Sciences, Zhengzhou University, Zhengzhou 450001, China

**Keywords:** cytoplasmic male sterility (CMS), mitochondrial genome, cotton (*Gossypium hirsutum*), full-length transcriptome, CMS-associated gene

## Abstract

As the core of heterosis utilization, cytoplasmic male sterility (CMS) has been widely used in hybrid seed production. Previous studies have shown that CMS is always closely related to the altered programming of mitochondrial genes. To explore candidate CMS genes in cotton (*Gossypium hirsutum*), sequencing and de novo assembly were performed on the mitochondrial genome of the *G. hirsutum* CMS line SI3A, with *G. harknessii* CMS-D2 cytoplasm, and the corresponding *G. hirsutum* restorer line 0-613-2R. Remarkable variations in genome structure and gene transcripts were detected. The mitochondrial genome of SI3A has three circle molecules, including one main circle and two sub-circles, while 0-613-2R only has one. RNA-seq and RT-qPCR analysis proved that *orf606a* and *orf109a*, which have a chimeric structure and transmembrane domain, were highly expressed in abortive anthers of SI3A. In addition, comparative analysis of RNA-seq and full-length transcripts revealed the complex I gene *nad4* to be expressed at a lower level in SI3A than in its restorer and that it featured an intron retention splicing pattern. These two novel chimeric ORFs and *nad4* are potential candidates that confer CMS character in SI3A. This study provides new insight into the molecular basis of the nuclear–cytoplasmic interaction mechanism, and that putative CMS genes might be important sources for future precise design cross-breeding of cotton.

## 1. Introduction

Cytoplasmic male sterility (CMS), a maternally inherited trait that inhibits efficient pollen production through a nuclear–cytoplasmic interaction has been widely exploited in the hybrid breeding of eudicot plants such as vegetable brassica, pepper, eggplant, tomato, fennel, carrot, and cotton [1,2,3,4,5,6,7,8]. In particular, CMS lines are usually employed as the female parents in hybridization, thus avoiding self-pollination, which produces non-hybrid seeds. Thus, CMS is of great agronomic importance. Mitochondria are semi-autonomous organelles with their own independent genome, the majority of genes in which are related to the complex subunit of the electron transport chain [9]. Previous studies in higher plants have shown that the CMS phenomenon is usually caused by aberrant recombination of the mitochondrial genome that leads to a highly-expressed chimeric open reading frame (ORF). For example, *orfH79* in CMS-HL rice is chimeric with *cox1* and an unknown sequence [10], *WA352* in CMS-WA rice is composed of three chimeric structures [11], and the chimeric gene, *atp6c,* in CMS-C maize contains a 481-bp leader sequence that is completely different from wild-type *atp6* [12]. It is generally believed that CMS chimeric genes are co-transcribed with genes involved in mitochondrial function, and furthermore, compete with or even inhibit the expression of genuine genes. For instance, *orf79* in CMS-BT is homologous with *orfH79* and co-transcribed with *atp6*, and *orf108* in *Brassica juncea* is co-transcribed with *atpA*, resulting in the CMS phenomenon [13,14,15]. Most CMS chimeric ORFs contain one or more transmembrane domains, such as *orf288* in *Brassica juncea* CMS-*hau*, *atp1-orf522* in CMS-PET1 sunflower, and *orf79* and *orfH79* in CMS-BT and CMS-HL rice [10,14,15,16,17]. In some cases, the mitochondrial retrograde regulation of nuclear genes causes pollen abortion. For example, in CMS-S maize, the nuclear-encoded gene, *ZmDREB1.7*, which is induced by mitochondrial retrograde signaling, promotes *orf355* accumulation that leads to subsequent male sterility [18]. Currently, the main strategy for identifying CMS genes is to compare CMS lines and their restorer lines at the mitochondrial genome and transcriptome levels.

Cotton is an important cash crop worldwide that provides natural fiber and vegetable oil. As early as 1894, heterosis was reported in the progeny of an interspecific hybrid between *Gossypium hirsutum* and *G. barbadense* [19]. Subsequently, more studies demonstrated heterosis performance in cotton to relate to improvements in fiber yield, fiber quality, disease resistance, and stress resistance [20,21,22]. Along with studies examining heterosis, research on cytoplasmic male sterility has also been carried out in cotton. The study of CMS in cotton began in the United States, mainly through distant hybridization and nuclear replacement with the goal of obtaining cytoplasmic male sterility. Since 1965, many cytoplasmic male-sterile lines have been cultivated across the world from the germplasm of *G. harknessii*, *G. arboreum*, *G. anomalum*, *G. tuilobum*, and *G. barbadense*. CMS lines of *G. hirsutum*, including 104-7A and Zhongmiansuo 12-A, have also been obtained by backcross hybridization in China [23]. To uncover the molecular mechanism of CMS in cotton, Wang et al. analyzed mitochondrial DNA from multiple male-sterile lines using the RAPD technology and determined the main driver of CMS in cotton to be the abnormality of the mitochondrial DNA [24]. Using the RFLP method, Feng et al. compared the mitochondrial genomes of CMS *G. harknessii* and normal fertile *G. hirsutum* and identified significant differences [25]. Wang et al. further analyzed the mitochondrial proteins and DNA from the anthers and etiolated seedlings of a CMS line and the corresponding restorer line of *G. harknessii* by SDS-PAGE, RAPD, and RFLP. They identified a 31 kDa polypeptide present in the anther mitochondria of the CMS line at the abortion stage. The RFLP analysis also showed that the mitochondrial DNA of the CMS line lacked a 1.9 kb fragment that has homology with the *coxII* gene that was present in the restorer line [26]. Li et al. compared the sterile line, 2074A and 2074S, as well as their restorer and maintainer lines and revealed four chimeric ORFs that were differentially transcribed in 2074A [27]. Khan et al. analyzed mitochondrial transcripts in the cotton CMS line, H276A, and identified a protein-coding gene, *cox3,* for which expression was lower than in the corresponding maintainer line H276B [28]. You et al. integrated methylome and transcriptome analyses for *G. barbadense* and revealed five key genes that may be associated with CMS [29]. However, although these experimental results have proven that CMS lines and restorer lines differ at the level of mitochondrial DNA, so far, no specific ORF with a chimeric structure similar to those implicated in the sterility of other plants has been identified in the mitochondrial genomes of CMS cotton lines.

In this study, we first reported the assembly of two complete mitochondrial genome sequences for the cotton CMS line, SI3A, and its restorer line, 0-613-2R, and further sequenced their full-length transcriptomes. The restoring line 0-613-2R completely restored fertility of the CMS line in the *G. harknessii* cytoplasm. A comparative analysis of the mitochondrial genomes and transcriptomes revealed differences in the sequences and expression patterns between the CMS and restorer lines and yielded several CMS-associated predicted chimeric ORFs that may be responsible for male sterility in SI3A. All told, the assembled cotton mitochondrial genome, the full-length transcriptome, and the set of candidate CMS genes in SI3A will contribute to our understanding of the molecular mechanism of CMS in cotton.

## 2. Results

### 2.1. Flower Morphology of the CMS-D2 Line SI3A and Its Restorer Line 0-613-2R 

Plants of the *G. harknessii* CMS-D2 cytoplasm line, SI3A, grew normally, and no morphological differences in vegetative growth were observed between SI3A and its restorer line, 0-613-2R. During the flowering period, the anthers of 0-613-2R were plump and the ovary enlarged, while in SI3A the stigma was elongated, the filaments were significantly shorter, and the anther sacs were completely wrinkled and did not dehisce (Figure 1A–D). The pollen grains of 0-613-2R were stained darkly by the I_2_-KI solution, while no pollen grains were stained for SI3A (Figure 1E,F). These findings prove that the CMS line, SI3A, is of the complete stamen degeneration type.

### 2.2. Mitochondrial Genome Assembly and Annotation 

The mitochondrial genomes of the CMS-D2 line, SI3A, and its restorer line, 0-613-2R, were sequenced with average depths of 167× and 312× using Illumina Hiseq and Nanopore sequencing technology. In the Illumina sequencing, 44.34- and 40.35-million clean reads were obtained from 0-613-2R and SI3A, respectively, and the corresponding Q20 values were 97.42% and 97.39%. In the Nanopore sequencing, 2.6- and 2.8-million clean reads were obtained from 0-613-2R and SI3A with total lengths of 14,071 Mb and 14,224 Mb, respectively; the lengths obtained for scaffold N50 were 8199 bp and 7142 bp. To verify the accuracy of the sequencing results, both Illumina and Nanopore sequencing reads were mapped to the cotton *G. hirsutum* genome (GCA_007990345.1) and the mitochondrial genome (NC_027406.1). Appendix A gives the average percentage of cytoplasm DNA (ctDNA) and nuclear DNA (ncDNA) identified in the Illumina and Nanopore sequence reads. Collinearity analysis revealed the SI3A and 0-613-2R mitochondrial genomes to be highly conserved with *G. hirsutum* (NC_027406.1) and *G. harknessii* (NC_027407.1); the 95.66% and 94.96% sequence identity (Figure 2) further indicates the conservation of the two mitochondrial genomes. 

The features of the mitochondrial genomes of the SI3A CMS line and its restorer line 0-613-2R are listed in Table 1. The SI3A mtDNA exhibited a more complex composition than that of the restorer line. Namely, the SI3A mtDNA consisted of one main genomic circle (SI3Amt2, 396,206 bp) and two sub-circle molecules (SI3Amt1, 63,927 bp; SI3Amt3, 174,070 bp). In contrast, the 0-613-2R mitochondrial genome assembled into a single circular molecule with a size of 607,367 bp. The guanine and cytosine (GC) content of the two mitochondrial genomes was approximately 45%, which is close to the median values of other fully-sequenced mitochondrial genomes from seed plants, such as 45.08% in *Brassica juncea*, 44.3% in wheat, 44.8% in *Arabidopsis thaliana*, and 45% in soybean [30,31,32,33]. 

Mitochondrial genes were annotated using the Mitofy software and compared against the protein-coding sequences from closely-related species to locate encoding regions. Within the SI3A mtDNA, a total of 33 protein-coding genes, 6 rRNAs, 25 tRNAs, and 59 unidentified ORFs were annotated, while the 0-613-2R mtDNA yielded annotations for 33 protein-coding genes, 6 rRNAs, 27 tRNAs, and 48 unidentified ORFs (Figure 3A,B). Notably, the two mitochondrial genomes were identical in terms of protein-coding gene number, while SI3A differed from 0-613-2R in having the *atp1* gene split into SI3Amt1 (190 bp) and SI3Amt3 (1335 bp), as well as lacking 87 bp from the end of *atp9*. Overall, protein-coding regions accounted for 25.8%, 19.0%, 7.4%, and 8.4% of the 0-613-2Rmt, SI3Amt1, SI3Amt2, and SI3Amt3 molecules, respectively. Detailed information describing the tRNA and rRNA gene content of the CMS and restorer line mitotypes is given in Appendix A. Interestingly, although the mitochondrial genome of SI3A was 26.8 Kb larger overall than that of the restorer line, 0-613-2R, it also featured fewer large repeats (>1 Kb). The presumed reason for this is that the repeated sequence that mediated gene rearrangements in the sterile line SI3A during assembly of the mitochondrial genome created its three circular DNA molecules, and thereby reduced the number of repeat sequences. 

### 2.3. Comparative Analysis of the SI3A and 0-613-2R Mitochondrial Genomes

#### 2.3.1. Repeat Sequences 

CMS in plants is maternally inherited and is generally controlled by mitochondrial genes. In most cases of CMS, rearrangements in the mitochondrial genome that are mediated by repeat sequences produce a new chimeric ORF that encodes proteins containing transmembrane domains [34,35,36]. Repeated sequences of greater than 50 bp annotated in the mitochondrial genomes of the SI3A CMS line and its restorer line are presented in Table 2 and Appendix A. Almost threefold as many repeats were detected in the restorer line than in the CMS line, at 77 repeats (77,020 bp, 12.8%) and 20 repeats (26,460 bp, 4.3%), respectively. All 20 repeats in SI3A occurred in SI3Amt2, the largest circular component of its mitochondrial genome; some shorter repeats (<50 bp) were present in the sub-genomes, SI3Amt1 and SI3Amt3. Five inverted repeats (2000 bp < IR > 1000 bp) were detected in 0-613-2R, all of which were absent from SI3Amt2 (Table 2). Two large repeats (>10 kb) were also identified in 0-613-2R, comprising one invert repeat (IR) and one direct repeat (DR) (Appendix A); only one large DR (>10 kb) was found in SI3Amt2, which showed 99% identity with those in the restorer line. The three large repeats also contained a few small repeats (<100 bp). Some sequences had multiple repeats with different loci in the two mitochondrial genomes. 

Notably, although the CMS line harbored fewer repeat sequences than the restorer line, and no long repeats (>2000 bp), direct repeats accounted for 50% of its repetitive sequences, twofold the proportion in the restorer line. This phenomenon may be related to gene recombination. Homologous recombination across large direct repeats separates the genome into sub-genomic molecules of different sizes, while recombination of invert repeats causes a sequence insertion [37]. The significant reduction of repeat sequences in the CMS line may be due to intragenomic homologous recombination of the largest direct repeat (10,637 bp), especially at non-coding sequences within the gene region, thereby causing the splitting of the SI3A mitochondrial genome into one main circle and two sub-circle structures.

#### 2.3.2. Structural Variation, SNPs, and InDels 

Genome structural variations, SNPs, and InDels between CMS line SI3A and its restorer line, 0-613-2R, were identified in order to explore structural and sequence variations in known mitochondrial genes and the predicted ORFs. In terms of structural variation, a total of 15 translocations, 1 inversion, and 11 translocation + inversion regions were identified (Figure 4A). Two insertions longer than 100 bp were also identified in syntenic regions, and 5 complex-InDel regions were detected. 

With regard to sequence variation, a total of 714 SNPs and 138 InDels (73 insertions and 65 deletions) were detected between the 2 mitochondrial genomes. The vast majority of identified SNPs occurred in non-coding regions, of which 700 (98% of the total SNPs) were distributed in intergenic regions, and only 14 (2% of the total SNPs) were identified in known mitochondrial genes, including *cox1*, *cox3*, *nad7*, *atp4*, *atp8*, *sdh3*, *matR*, *rps4*, *rpl2*, *rpl5*, *rpl10*, and *rpl16.* Five were synonymous mutations, and not expected to cause the CMS, while nine were non-synonymous mutations that altered the encoded amino acid residues in nine mitochondrial genes (*cox3*, *atp8*, *sdh3*, *matR*, *rps4*, *rpl2*, *rpl5*, and *rpl10*) (Table 3). These non-synonymous changes are candidates for functional scrutiny in investigating the molecular mechanisms of CMS, since protein-coding genes in plant mtDNA are extraordinarily conserved and their rate of evolution is very low [38,39,40]. These SNPs primarily concerned A and C bases, and most were actually transversions rather than transitions. 

The identified InDels had lengths in the range of 1–10 bp, with almost all being 1 bp (Figure 4B). Most were sited in intergenic regions; one insertion of a “T” in the coding region of SI3A *atp1* resulted in a frameshift mutation affecting 44 amino acids (Figure 4C). Mutations in *atp1* may impair ATP synthesis in the mitochondrial respiratory chain. No other InDels resulted in a frameshift.

### 2.4. Unique ORFs 

To date, CMS-associated genes have been mainly identified by whole-genome sequencing of mitochondria and subsequent comparison with a standard cultivar to identify unique ORFs. In this study, 30 and 38 unique ORFs encoding >100 amino acids were identified in the 0-613-2R restorer line and the SI3A CMS line. respectively (Appendix A). Moreover, 22 specific ORFs in SI3A were determined to be distinct from ORFs in 0-613-2R (Figure 5A). CMS-associated genes are known to be characterized by chimeric structures or to be co-transcribed with functional genes, and said functional genes usually contain transmembrane domains. Here, only two ORFs were predicted to specifically exist in the CMS line, have a chimeric structure, and be co-transcribed with genes containing transmembrane domains. In addition, at least one transmembrane domain was found in these two ORFs. The first ORF, *orf119a-2*, shared 73 bp of sequence with the known gene *nad7* and also featured 284 bp of unknown sequence, while the second, *orf606a*, shared 1336 bp at its 5′ end with the known gene *atp1* and contained another 485 bp of unique sequence (Figure 5B). We also chose to examine two other unique ORFs identified in SI3A, *orf131a* and *orf109a*, which featured chimeric structures. Of those, *orf131a* contained a 92 bp sequence corresponding to the 3′ end of *nad6* plus 304 bp of unknown sequence, while *orf109a* fused 35 bp corresponding to the 3′ end of *cox2* along with a 291 bp sequence of unknown origin (Figure 5B). Since having a transmembrane domain is another characteristic of CMS genes, we performed a structural analysis of the four selected ORFs; this revealed that *orf119a-2* and *orf606a* each contain a transmembrane structural domain, while *orf131a* and *orf109a* do not (Figure 5C–F).

### 2.5. Selection of Candidate Genes Resulting in Male Sterility of SI3A

To further explore the expression of CMS-associated genes and their intron splicing patterns in SI3A, Illumina and full-length PacBio transcriptome sequencing were performed. Specifically, RNA from abortion stage tissue (4–5 mm floral bud) of CMS line SI3A and its restorer line, 0-613-2R, was sequenced using the Illumina NovaSeq 6000 platform and PacBio Sequel II System. PacBio sequencing yielded a total of 42.2 M and 28.0 M subreads from 0-613-2R and SI3A, and the length of scaffold N50 was 2111 bp and 2122 bp, respectively. For Illumina RNA-Seq, 150-bp paired-reads were generated and the Q20 percentages exceeded 97% in each sample. In total, 82.2 Mb and 77 Mb of raw reads were obtained for 0-613-2R and SI3A, respectively, corresponding to 78.8 Mb and 73.5 Mb of clean reads after filtering. Clean reads were mapped to the two mitochondrial genomes using DEG-seq2 for a transcript-level examination of mitochondrial protein-coding genes (Table 4). Novel transcript prediction subsequently identified 99 and 91 novel transcripts in SI3A and 0-613-2R, respectively, including 34 and 33 protein-coding transcripts. Of protein-coding genes in SI3A, we found *rps10* to be the most highly expressed and *nad4* the least expressed (Figure 6A). In addition to quantifying expression levels from the RNA-seq data and visual inspection in IGV, RT-PCR analysis was performed for the four ORFs in abortion floral buds from SI3A and 0-613-2R. The ORFs did not show obvious differences between the two lines except for *orf606a*, which was significantly more highly expressed in SI3A (Figure 6C). We also analyzed relative gene expression using RT-qPCR and found that *orf109a* and especially *orf606a* are highly expressed in SI3A (Figure 6B), which is consistent with the RT-PCR results. Sequence comparison revealed *orf606a* to be partially identical to *orf610a*, which was previously reported as the CMS candidate gene in cotton; in particular, a transgenic *Arabidopsis thaliana* line harboring *orf610a* with a mitochondrial signal peptide exhibits partial pollen abortion, shortened siliques, shortened filaments, and pollen tubes [41]. Accordingly, *orf606a* is a candidate sterility gene for validation in future work using transgenic cotton.

The present work identified a total of seven intron-containing protein-coding genes in the two mitochondrial genomes. Full-length PacBio sequencing revealed SI3A to exhibit an intron retention splicing pattern for *nad4*, in which intron 1 was almost completely preserved and intron 3 was partially preserved (Figure 6D); corresponding transcripts were detected in RNA-seq data from SI3A but not in data from 0-613-2R. Therefore, the differential transcript caused by intron retention may be related to the low expression of *nad4* in the CMS. 

## 3. Discussion

### 3.1. Characteristics of the SI3A and 0-613-2R Mitochondrial Genomes

The complete mitochondrial genomes of CMS line SI3A and the corresponding restorer line, 0-613-2R, were constructed through sequencing and de novo assembly. The mtDNA sequences of SI3A and 0-613-2R were nearly identical to those of the 2074A and 2074S *G. hirsutum* CMS lines for which mitochondrial genomes were previously reported [27]. The differences in mitochondrial genome size between CMS and restorer lines may be caused by unique sequences, as has been observed in tomato and wheat [32,42]. Here, unique sequences in SI3Amt2 (21.3 kb) and SI3Amt3 (1.1 kb) were the main factors leading to the size difference between SI3A and 0-613-2R. Notably, high-frequency homologous recombination mediated by large repeat sequences results in sub-circle structures, such as those observed in SI3A. Higher plant mitochondrial genomes are frequently composed of a main circle structure and multiple sub-genomic molecules [43]; for example, the mitochondrial genome of monkeyflower (*Mimulus guttatus*) contains three large repeats for which four possible recombination relationships have been deduced, ultimately producing eight sub-ring structures [44]. In soybean, the CMS-K line, NJCMS1B, features a main circle, a linear sequence, and a sub-genomic circle [31]. Here, one and two large (>10 kb) repeat sequences were identified in SI3A and 0-613-2R, respectively (Table 2 and Appendix A). Intra-genomic homologous recombination of the large repeat sequence in SI3A resulted in its main circle and two sub-circle structures (Figure 3B). Notably, SI3A exhibited a larger genome but significantly fewer repeats than 0-613-2R, which may be due to the increased frequency of gene rearrangements caused by the generation of the sub-circle structures, ultimately disrupting the original repeat sequences. 

Functional mitochondrial genes are, in general, quite conserved. Between-species differences in the number of mitochondrial coding genes are mainly caused by multi-copy genes, ribosomal RNA, and tRNA [45]. Here, both examined genomes harbored 33 protein-coding genes (Table 1), the finding of which is consistent with 2074A and 2074S—except for the loss of *nad1* and *nad2* [27]. This discrepancy may be related to the sequencing depth and annotation method. Additionally, a comparative analysis revealed nine non-synonymous SNP mutations between SI3A and 0-613-2R, which occurred in eight protein-coding genes (*cox3*, *atp8*, *sdh3*, *matR*, *rps4*, *rpl2*, *rpl5*, and *rpl10*), along with just one exonic InDel site, which caused a frameshift mutation in *atp1* (Table 3 and Figure 4A,B). Mutations in *atp1* may disrupt the assembly of complex V, and hence ATP synthesis in the mitochondrial electron transport chain. 

### 3.2. orf606a and nad4 May Be CMS Candidate Genes for the CMS-D2 Line SI3A

CMS genes are always associated with novel chimeric ORFs created by mitogenome recombination and are often co-transcribed with upstream or downstream protein-coding genes. For example, in *Raphanus sativus* DCGMS, *orf463* represents a partial fusion of *cox1* with an unidentified sequence [46], while in CMS-S maize, *orf77* contains three segments derived from *atp9* [47]. This study identified four ORFs specific to SI3A, *orf109a*, *orf119a-2*, *orf606a* and *orf131a* (Figure 5B), which were all chimeric with protein-coding genes and contained one or more transmembrane domains (Figure 5C–F). Of those, *orf606a*, which was chimeric with *atp1* and an unknown sequence, was found to be especially highly expressed in abortion floral buds of SI3A compared with 0-613-2R (Figure 6B,C). In addition, the sequence of *orf606a* is homologous with the CMS gene, *orf610a*, previously reported in the CMS-D2 line ZBA, which causes ROS accumulation and ATP reduction [41], and its expression pattern is consistent with other known CMS genes, such as *orf256* in wheat, *cox3* in H276A cotton, and *orf326* in Nau radish [28,48,49].

Most reported restorer-of-fertility (*Rf*) genes that counteract male sterility encode members of the pentatricopeptide repeat (PPR) family; these include petunia *Rf* [50], *Brassica napus Rfk1* [51], *Rahanus stivus Rfo* [52], rice *Rf1* [53], sorghum *Rf1* [54], and others. Most PPR proteins localize to the mitochondria and chloroplasts and play important roles in post-transcriptional regulation within those organelles, mainly in RNA splicing, RNA cleavage, RNA editing, and RNA stabilization [55,56]. For example, the PPR protein, Rf6, is known to function with OsHXK6 to promote processing of the *atp6–orfH79* transcript and restore fertility in rice [57], while *OTP43* is essential for *trans*-splicing of *nad1* intron 1 in *Arabidopsis thaliana* [58]. In this study, RNA-seq and full-length transcript data revealed *nad4* to have low expression in SI3A, and furthermore that retention of intron 1 resulted in the production of abnormal transcripts (Figure 6A,D). In the current literature, most known PPR proteins are involved in *cis–trans*-splicing of *nad1*, *nad2*, and *nad4*. For example, the PPR protein, MTSF3, is involved in 3’ end processing and intron *trans*-splicing of mitochondrial *nad2* mRNA [59]. Similarly, in CMS-S maize, PPRK2 (Rf3) targets mitochondria, binds to *orf355*, and inhibits its editing and degradation to restore fertility [60]. The PPR protein, Ghlm, has likewise been demonstrated to be necessary for splicing of *nad7* mRNA during cotton fiber development [61]. In light of these reports, we speculate that the abnormal transcript generated from *nad4* might be the target of the cotton restorer protein, Rf1. Thus, *orf606a* and *nad4* may be potential candidate CMS-associated genes in cotton. We hope these results will contribute to increasing our understanding of the cause of CMS in cotton and provide candidate CMS genes for hybrid breeding.

## 4. Materials and Methods

### 4.1. Plant Materials and Phenotypic Analysis

The plant materials used in this study were the *Gossypium hirsutum* nuclear restorer line, 0-613-2R, with viable pollen, and SI3A, an upland CMS-D2 line with cytoplasm from *G. harknessii*. Plants were grown under standard growing conditions in the experimental field of Zhejiang University (30.1° N, 120.1° E), Hangzhou, in which the average daily temperature ranged from 30 °C to 35 °C. Floral buds of about 4–5 mm in length (the abortion stage) from 100 plants, each of CMS-D2 line SI3A and its restorer line 0-613-2R, were collected and combined for RNA extraction. All harvested samples were frozen in liquid nitrogen and stored at −80 °C before use. Flower morphology was observed by stereoscopic microscopy (Stemi 508, Suzhou, China). Pollen viability was evaluated on the flowering day by staining with 1% I2-KI solution after crushing anthers, with the staining observed under a microscope (XSP-37XC, Shanghai, China).

### 4.2. Library Construction, Sequencing, and Assembly of Mitochondrial Genomes

High-quality DNA was extracted from leaves using a modified CTAB method [62]. The purity, concentration, and integrity of the DNA samples were confirmed by Nanodrop (OD260/280 >1.8) and agarose gel electrophoresis. Afterwards, the purified DNA was fragmented with a Covaris instrument and used to construct a paired-end library with an average fragment length of 350 bp using the NexteraXT DNA Library Preparation Kit (Illumina, San Diego, CA, USA). Sequencing was then performed on the Illumina Novaseq 6000 platform (Huitong Biotechnology Co. Ltd., Shenzhen, China). In parallel, fragmentation was performed using a BluePippin system (Sage Science, Beverly, MA, USA), then a single-end library (average length 8 kb), was constructed using the SQK-LSK109 kit (Oxford Nanopore, Oxford, UK) following the manufacturer’s instructions, and was sequenced on the Nanopore PromethION sequencing platform (Huitong biotechnology Co. Ltd., Shenzhen, China).

Before assembly, the original Illumina reads were filtered to remove reads with adaptors, reads showing a quality score below 20 (Q < 20), short fragments of less than 50 bp after adapter removal, and reads containing greater than 10% uncalled bases (“N” characters). The mitochondrial genome was reconstructed using the Oxford Nanopore data and Illumina NovaSeq data in combination as follows. First, the clean Illumina sequencing data were de novo assembled by SPAdes v3.11.0 (Bankevich A, Petersburg, Russia) using default parameters without a cutoff. Second, the scaffolds were filtered using BlastN and Exnerate with an alignment threshold of e-value 1e-10 and protein similarity threshold of 70%. Selected scaffolds having homology to the *Gossypium arboreum* (NCBI: KR736342) mitochondrial genome and protein-coding gene sequences were removed as fragments that were low coverage and obviously not the target genome. The collected fragmented target sequences were extended and merged using PRICE and MITObim [63] to minimize the number of scaffolds. Then, the Nanopore reads were aligned back to our filtered SI3A and 0-613-2R scaffolds with minimap2 [64], segregated by aligned reads, and reassembled *de novo* with Flye [65]. The final genome sequences of the CMS-D2 line, SI3A, and its restorer line, 0-613-2R, were obtained by polishing with pilon [66] using Illumina sequencing reads and made publicly available through the National Center for Biotechnology Information (NCBI) under the GenBank accession numbers: SI3Amt1, ON931349; SI3Amt2, ON931350; SI3Amt3, ON931351; and 0-613-2R, ON931352.

### 4.3. Mitochondrial Genome Annotation and Identification of ORFs

The mitochondrial genes were annotated by Motify [67]. Protein sequences were aligned with the cotton mitochondrial reference genomes from *Gossypium arboreum* (NCBI: KR736342), *G. hirsutum* (NC_027406.1), and *G. harknessii* (NC_027407.1), and the resulting gene sets were then integrated and manually corrected to obtain the final list of protein-coding genes. Transfer RNA (tRNA) and ribosomal RNA (rRNA) genes were predicted using the homology prediction methods of tRNAscan-E and rRNAmmer [68]. Gene function annotation was carried out using BLAST with e-value 1 × 10^−10^ and the following databases: Kyoto Encylopedia of Genes and Genomes (KEGG), Clusters of Orthologous Groups (COG), Swiss-Prot, Gene Ontology (GO), and the NCBI non-redundant protein sequences (NR). ORFs were predicted with the NCBI ORF Finder (https://www.ncbi.nlm.nih.gov/orffinder/) (accessed on 9 April 2022) and retained if the hypothetical protein encoded by the ORF was longer than 100 amino acids. ORFs were named according to the number of amino acids encoded. Hypothetical protein transmembrane domains were predicted by the TMHMM online server v2.0 http://www.cbs.dtu.dk/services/TMHMM/ (accessed on 6 May 2022).

### 4.4. Differential Analysis of the SI3A and 0-613-2R Mitochondrial Genomes

Using restorer line 0-613-2R as the reference genome, syntenic analysis was performed to compare the sequence information of these two mitochondrial genomes in detail. First, large-scale synteny between the genomes was determined by MUMmer 4.0 (Alekley Zimin, Baltimore, MD, USA) [69]; then, LASTZ https://lastz.github.io/lastz/ (accessed on 31 September 2020) was used to compare blocks and confirm the local position alignment. MUMmer 4.0 and BLAST were employed for global alignment to identify differential single nucleotide polymorphism (SNP) loci. InDel sequences were obtained from the LASTZ alignment and verified by BWA [70,71] and SAMtools [72] filtering to arrive at a reliable set of InDels. Repeat sequences were predicted using RepeatMasker 4.1.1 https://www.repeatmasker.org/ (accessed on 15 October 2020).

### 4.5. RNA Extraction and Transcriptome Sequencing

Total RNA was extracted from floral buds (4–5 mm size, abortion stage) of SI3A and 0-613-2R using the Spectrum Plant Total RNA Kit (Sigma-Aldrich, St. Louis, MO, USA) with three replicates for each sample. RNA concentration and integrity were assessed with the Fragment Analyzer 5400 (Agilent Technologies, CA, USA). Sequencing libraries were generated using the NEBNext Ultra TM RNA Library Prep Kit for Illumina (NEB, Ipswich, MA, USA), and index codes were added in order to attribute sequences to each sample. Briefly, mRNA was purified from total RNA using poly-T oligo-attached magnetic beads and a fragmentation reaction was carried out under elevated temperature to break the mRNA into short template pieces. Second-strand cDNA synthesis was subsequently performed using DNA polymerase I and RNA H, after which the library fragments were purified using the AMPure XP system (Beckman Coulter, Beverly, MA, USA). After adaptor-ligase and end repair reactions, suitable fragments were selected for PCR amplification, the resulting PCR products were purified (AMPure XP system) (Beckman Coulter, Beverly, MA, USA), and the library quality was assessed on the Agilent Bioanalyzer 2100 system. Clustering was then performed on a cBot Cluster Generation System using the TruSeq PE Cluster Kit v3-cBot-HS (Illumina, San Diego, CA, USA) according to the manufacturer’s instructions. After cluster generation, the prepared libraries were sequenced on an Illumina NovaSeq 6000 Platform, generating 150 bp paired-end reads. Raw reads were filtered by discarding any pair in which one read contained adapter contamination, more than 10% uncertain bases, or more than 50% low quality (Phred quality <5) bases. The clean sequence files of these six samples (FASTQ files) are accessible from the NCBI through the Sequence Read Archive (SRA) under accession number SRR20067583-SRR20067588.

### 4.6. Full-Length Transcriptome Sequencing

The total RNA from 4–5 mm floral buds was first assessed for quality by measuring RNA Integrity Number (RIN) and concentration using an Agilent 2100 instrument; then, 2 µg of total RNA was used as the input material for RNA sample preparation. The SMRTbellTM Template Prep Kit (Pacific Biosciences, Menlo Park, CA, USA) was used to generate SMRTbellTM libraries by following the manufacturer’s instructions for end-repair, A-tailing, and adaptor ligation, after which the libraries were sequenced on the PacBio Sequel II System (Pacific Biosciences, CA, USA). Sequence data were processed using the SMRT Analysis software (ISO-seq version 3.0) (Pacific Biosciences, CA, USA) with parameters: min length, 300; min accuracy, 0.75; and min passes, 0. Finally, high-quality isoforms and polished low-quality isoforms were mixed for subsequent alignment to the reference genome using GMAP [73,74] with parameters -min-trimmed-coverage 0.85 and -min-identity 0.9 against reference genome. Any redundancy in polished consensus reads was removed by TOFU (-min-coverage 0.85 and -min-identity 0.9 against reference genome) to obtain final transcripts for subsequent analysis.

### 4.7. RT-PCR and RT-qPCR Analysis of Candidate ORFs

Floral buds (4–5 mm, abortion stage) of the CMS line, SI3A, and its restorer line, 0-613-2R, were used in measuring the expression of candidate ORFs. First, the total RNA was extracted with the Spectrum Plant Total RNA Kit (Sigma-Aldrich, St. Louis, Mo, USA); then, reverse transcription was performed using the HiScript QRT SuperMix for RT-qPCR kit (Vazyme, Nanjing, China) following the manufacturer’s protocol. Semiquantitative RT-PCR was then performed with the following temperature program: 95 °C for 5 min, then 32 cycles of 95 °C for 15 s, 58 °C for 15 s, 72 °C for 30 s, and 5 min at 95 °C. RT-qPCR was performed with the ABI StepOnePlus system using AceQ RT-qPCR SYBR Green Master Mix (Vazyme, Nanjing, China). Values were standardized according to the smallest sample threshold cycle number (Ct value) and the highest fluorescence value, and relative gene expression levels were calculated using the 2^−ΔΔCT^ method [75]. *GhHis3* was used as the internal reference gene for RT-PCR and RT-qPCR. Appendix A lists the primers used in RT-PCR and RT-qPCR.

## Figures and Tables

**Figure 1 ijms-23-09198-f001:**
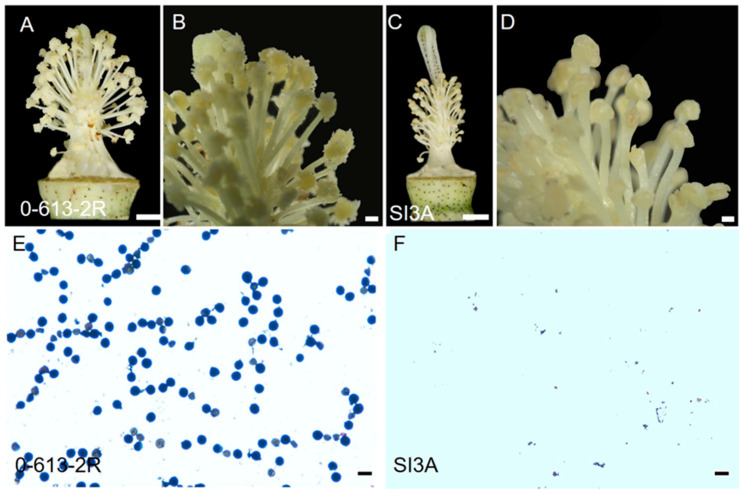
Floral organs morphology and pollen shed of the CMS line, SI3A, and its restorer line, 0-613-2R. (**A**,**C**), The stamens and pistils of restorer line 0-613-2R (fertile) and CMS line SI3A (sterile). (**B**,**D**) The enlargement of the anther. (**E**,**F**) Pollen grains stained with I2-KI. Scale bars correspond to 3 cm (**A**,**C**), 0.5 cm (**B**,**D**), and 100 μm (**E**,**F**).

**Figure 2 ijms-23-09198-f002:**
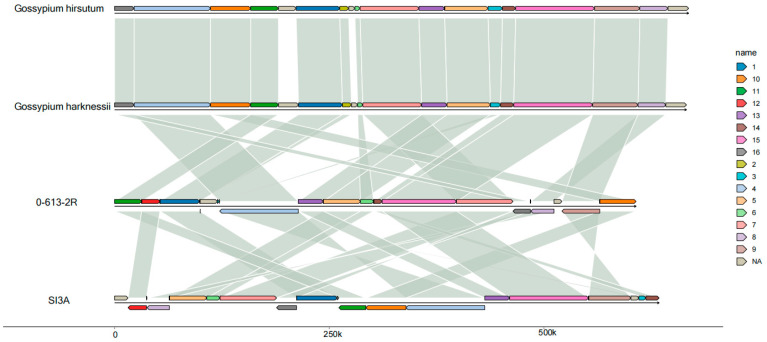
Sequence collinearity analysis of 0-613-2R and SI3A mitochondrial genome with *G. hirsutum* and *G. harknessii* mitochondrial genome. Different colors represent different blocks in each mitochondrial genome and the two blocks that are connected by corresponding line indicates high homology between them. Reverse and direct transcript orientation are indicated above and below the central line.

**Figure 3 ijms-23-09198-f003:**
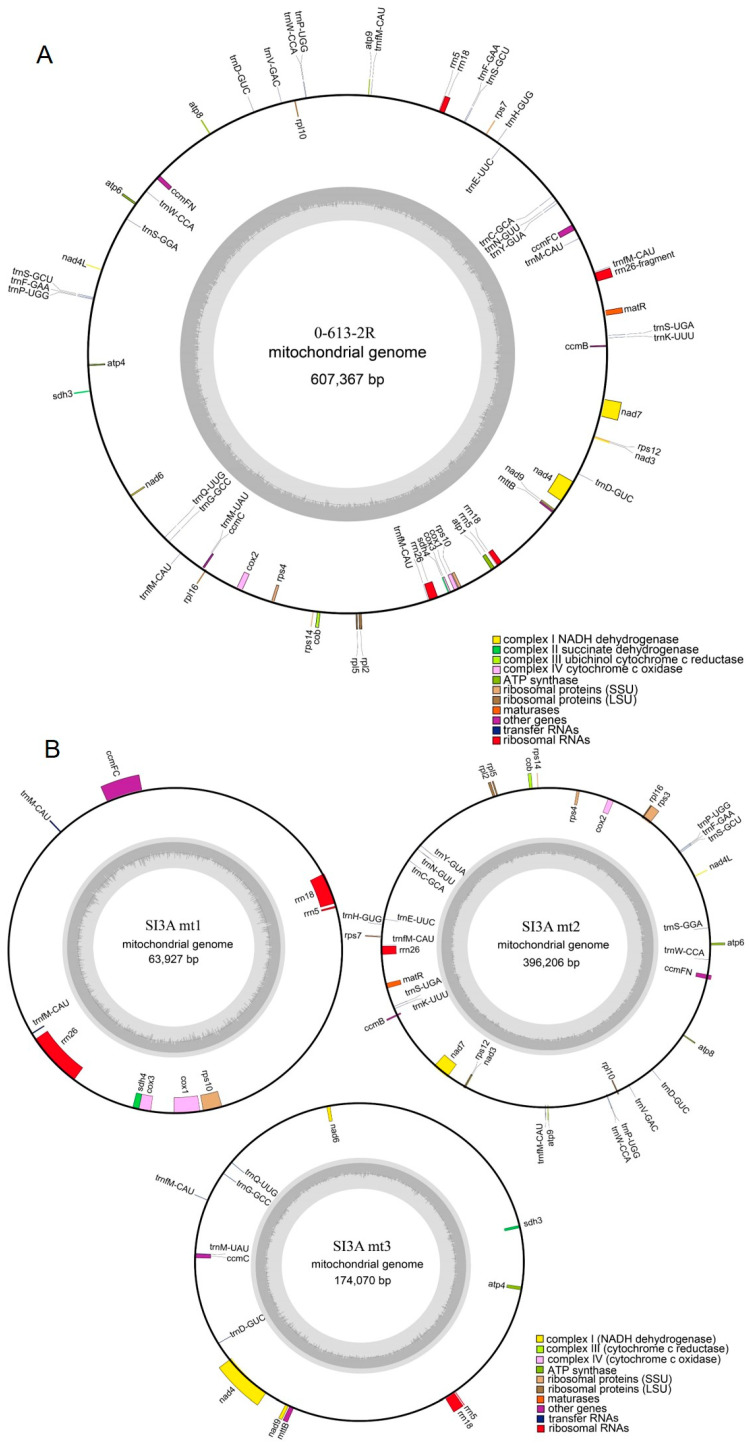
Mitochondrial genome map of the restorer line, 0-613-2R, and the CMS line, SI3A. (**A**) Mitochondrial genome map of restorer line, 0-613-2R. (**B**) Mitochondrial genome map of CMS line, SI3A. Genes inside the loop represent transcription in a clockwise direction, while genes outside the loop are in the opposite direction. Different functional genes are marked with different colors. The built-in gray histogram shows the genomic GC content, and the middle gray line is the 50% threshold line.

**Figure 4 ijms-23-09198-f004:**
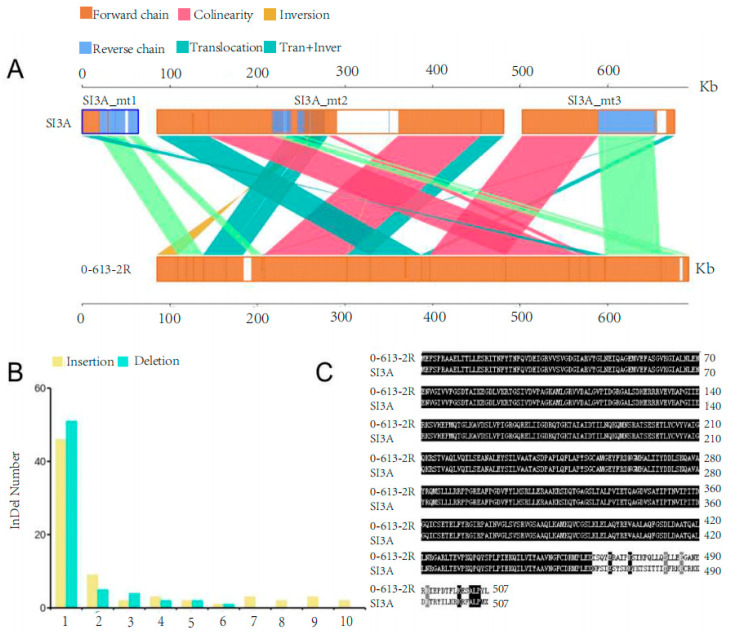
Structural variation and InDels number in mitochondrial genomes of SI3A and 0-613-2R. (**A**) The structural variation in SI3A and 0-613-2R. Forward chain: forward chain of the genome sequence; Reverse chain: reverse chain of the genome sequence; Collinearity: the same linear region; Tran + Inver: the area of translocation and inversion; Inversion: the area of inversion; Translocation: the area of translocation. (**B**) The length of InDels and its corresponding number in SI3A mitochondrial genome compared with 0-613-2R. Yellow, insertion; Blue, deletion. (**C**) Amino acid sequences alignment of *atp1* between 0-613-2R and SI3A.

**Figure 5 ijms-23-09198-f005:**
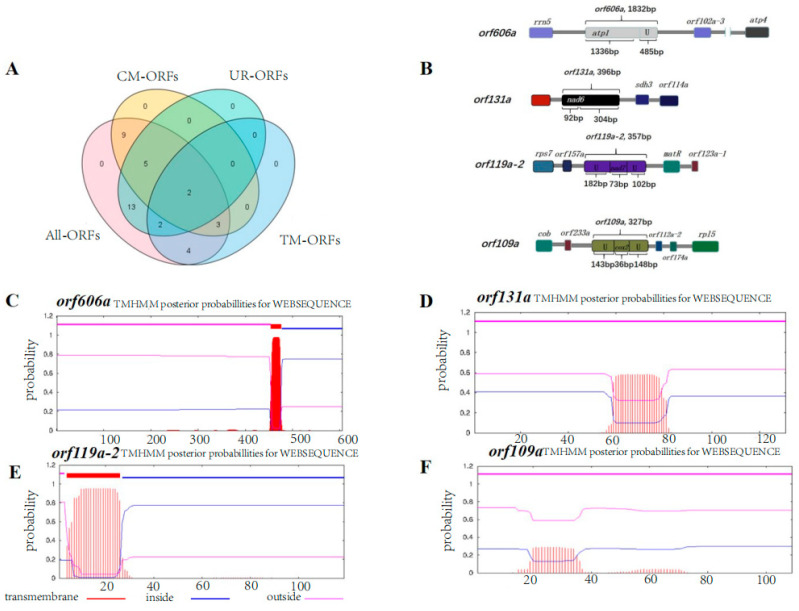
Distribution of four types of ORFs and chimeric structures, transmembrane domain of the specific ORF in SI3A. The Venn diagram (**A**) shows the distribution of four types of ORFs. All ORFs: numbers of all the detected ORFs in SI3A; CM-ORFs: numbers of chimeric structure ORFs in SI3A; UR-ORFs: numbers of unique region ORFs in SI3A; TM-ORFs: numbers of transmembrane domain ORFs in SI3A. (**B**) Chimeric structure of *orf606a*, *orf131a*, *orf119a-2*, and *orf109a*. Rectangle boxes represent coding sequences, different colors represent different coding genes, linear represents the different size of the noncoding sequences, and U represents the unknown sequences. (**C**–**F**) The location and probabilities of transmembrane domains of the four specific ORFs *orf606a* (**C**), *orf131a* (**D**), *orf119a-2* (**E**), and *orf109a* (**F**).

**Figure 6 ijms-23-09198-f006:**
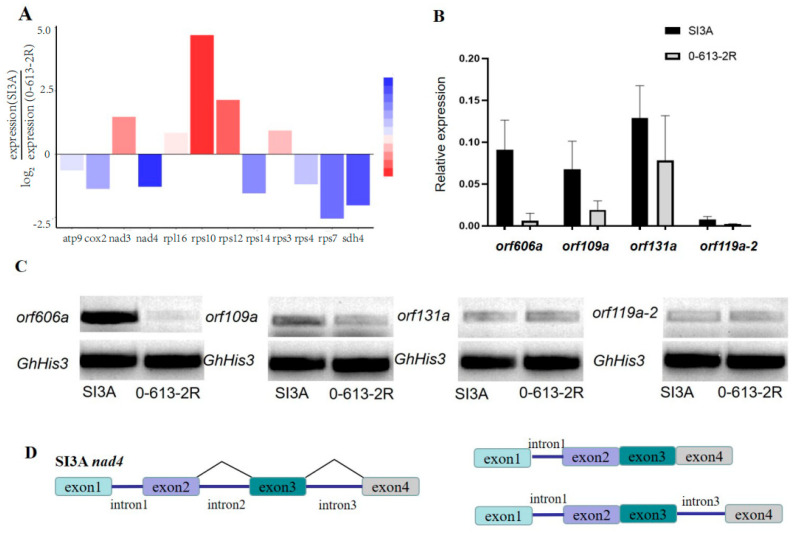
Different expression levels of mitochondrial genes and schematic diagram of *nad4* intron splicing patterns. (**A**) Relative expression levels of mitochondrial genes with significant differences in RNA−seq data. Blue represents significantly downregulated genes and red represents significantly upregulated genes in SI3A. (**B**) RT−qPCR analysis of four specific ORFs expression in CMS line SI3A and restorer line 0-613-2R, *GhHis3* was used as the internal control. (**C**) RT−PCR analysis of the four specific ORFs in 0-613-2R and SI3A. (**D**) Schematic diagram of *nad4* intron splicing patterns, two transcripts formed after splicing retention of intron 1 and intron 3 are shown on the right.

**Table 1 ijms-23-09198-t001:** Features of the assembly mitogenomes in the CMS line, SI3A, and its restorer line, 0-613-2R.

Genome Characteristics	0613-2R mt	SI3A		
		SI3A mt1	SI3A mt2	SI3A mt3
Genomic size(bp)	607,367	63,927	396,206	174,074
G + C content (%)	44	46	44	45
coding sequence (%)	25.3%	19.0%	7.4%	8.4%
ORF	48	12	33	14
Protein coding genes	33	6	20	7
tRNA genes	27	2	18	5
rRNA genes	6 ^1^	3 ^1^	1 ^1^	2 ^1^
Repeat content percent coverage of total genome	16.2%	0.8%	8.8%	9.2%
Large repeats: >1 kb(number)	13	0	5	0
Small repeats: <1 kb(number)	427	11	169	34

^1^ Multiple copies of rRNAs.

**Table 2 ijms-23-09198-t002:** Repeat sequences (>50 bp) in the mitogenomes of CMS line SI3Amt2.

		Copy1				Copy2			
No	Size (bp)	Start1	Stop1	Type	Size (bp)	Start2	Stop2	Identity (%)	Difference between Copies
AR1	10,637	120,585	131,221	DR ^1^	10,636	359,345	369,980	99	copy1 9 bp indel, copy2 11 bp indel.
AR2	435	237,794	238,228	DR	435	334,087	334,521	100	
AR3	333	211,333	211,665	IR ^2^	333	360,536	360,868	100	
AR4	229	154,125	154,353	DR	229	254,009	254,237	100	
AR5	203	237,590	237,792	DR	203	333,883	334,085	100	
AR6	175	83	257	IR	175	281,969	282,143	100	
AR7	154	56,027	56,180	DR	154	359,760	359,913	100	
AR8	121	56,027	56,147	DR	121	121,000	121,120	100	
AR9	113	40,192	40,304	DR	113	102,378	102,490	100	
AR10	113	57,141	57,253	DR	113	212,119	212,231	100	
AR11	107	201,247	201,353	DR	107	389,303	389,409	100	
AR12	93	111,326	111,418	DR	93	331,353	331,445	100	
AR13	78	75,293	75,370	IR	78	358,615	358,692	100	
AR14	71	264,143	264,213	IR	71	360,450	360,520	100	
AR15	66	75,146	75,211	IR	66	116,361	116,426	100	
AR16	66	211,392	211,457	IR	66	297,562	297,627	100	
AR17	66	297,562	297,627	DR	66	360,744	360,809	100	
AR18	60	40,306	40,365	DR	60	102,492	102,551	100	
AR19	57	7252	7308	DR	57	324,694	324,750	100	
AR20	53	47,233	47,285	IR	53	170,558	170,610	100	

^1^ DR: direct repeats, ^2^ IR: invert repeats.

**Table 3 ijms-23-09198-t003:** The annotation of SNP in the mitogenomes of SI3A compared with 0-613-2R.

Gene	Location	0-613-2R<->SI3A (Nucleic Acid)	0-613-2R<->SI3A (Amino Acid)	Mutation Type ^1^	SNP Type
cox1	495,890	C<->A	Ile<->Ile	S	transversion
cox1	496,358	C<->A	Ile<->Ile	S	transversion
cox3	494,152	C<->A	Leu<->Ile	N	transversion
nad7	583,680	A<->C	Ile<->Ile	S	transversion
atp4	307,826	C<->T	Phe<->Phe	S	transition
atp8	205,705	C<->A	Ser<->Arg	N	transversion
sdh3	317,242	A<->C	Leu<->Phe	N	transversion
matR	16,075	C<->A	Cys<->Lys	N	transversion
rps4	426,548	A<->C	Lys<->Cys	N	transversion
rpl2	458,973	T<->G	Phe<->Leu	N	transversion
rpl2	459,220	A<->C	IIe<->Leu	N	transversion
rpl5	457,988	A<->C	Lys<->Cys	N	transversion
rpl10	171,493	G<->A	Gln<->Lys	N	transition
rpl16	399,033	A<->C	Val<->Val	S	transversion

^1^ S: synonymous, N: non-synonymous.

**Table 4 ijms-23-09198-t004:** Mapping results for Illumina RNA-seq data to mitochondrial genome.

Sample	0-613-2R-1	0-613-2R-2	0-613-2R-3	SI3A-1	SI3A-2	SI3A-3
Total raw reads	27,570,398	27,645,300	27,163,449	24,396,971	30,170,978	22,451,704
Total clean reads	26,157,634	26,560,996	26,094,847	23,354,551	28,705,908	21,508,933
Clean reads Q20(%)	97.38	97.4	97.22	97.33	97.13	97.38
Clean reads Q30(%)	92.59	92.66	92.25	92.44	92.08	92.59
Total alignment with mt genome	25,765 (0.10%)	25,125 (0.09%)	26,713 (0.10%)	16,247 (0.07%)	21,734 (0.08%)	18,759 (0.09%)
Exact alignment with mt genome	14,708 (0.06%)	14,376 (0.05%)	15,446 (0.06%)	12,592 (0.05%)	16,988 (0.06%)	14,509 (0.07%)
Average (exact)	0.06%			0.06%		

## Data Availability

The mt genome of 0-613-2R and SI3A are available at NCBI GenBank, under the accession number: SI3Amt1, ON931349; SI3Amt2, ON931350; SI3Amt3, ON931351; 0-613-2R, ON931352. The Illumina RNA-seq data are available at the Sequence read Archive under accession number SRR20067583-SRR20067588.

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
