# Peer review of "Comparison of Mitochondrial Genomes between a Cytoplasmic Male-Sterile Line and Its Restorer Line for Identifying Candidate CMS Genes in Gossypium hirsutum"

_ijms, 2022, doi:10.3390/ijms23169198_

Round 1

Reviewer 1 Report

Xuan et al, produced a treasure-trove of molecular data by combining second and third generation sequencing analyses. The aim was to assemble the mtDNA of two lines (MF and CMS) of cotton and to combine both genomic and transcriptomic analysis in order to identify possible candidate genes responsible for CMS. Despite the potential value of this work, the manuscript is poorly written, several key-analysis are missing, the Methods section needs to be totally rewritten and some analyses are obscure. There isn’t any major finding nor a real candidate gene is proposed and further investigated. The English form must be deeply edited.

I'm confident that - after a strong round of revisions - the manuscript will be suitable for a re-consideration. Please ask for assistance for English form and for the bioinformatic section.

 Abstract

26-29 It is a repetition of what already reported in 17-19 and 23-24

Please cite within the abstract at least one time the species you are analyzing (G. hirsutum)

Introduction

38 This is not exhaustive: CMS is widely applied also in other species. Instead of making a broad description, (that includes monodicots and dicots), I would focus only on eudicots (such us cotton) by providing example of the main families where CMS is widely exploited:

- Brassicaceae (1) Singh, S.; Dey, S.S.; Bhatia, R.; Kumar, R.; Behera, T.K. Current understanding of male sterility systems in vegetable Brassicas 466 and their exploitation in hybrid breeding. Plant Reprod 2019, 32, 231–256, doi:10.1007/S00497-019-00371-Y. 467; 2) Singh, S.; Dey, S.S.; Bhatia, R.; Kumar, R.; Sharma, K.; Behera, T.K. Heterosis and combining ability in cytoplasmic male sterile 468 and doubled haploid based Brassica oleracea progenies and prediction of heterosis using microsatellites. PLoS One 2019, 14, 469 e0210772, doi:10.1371/JOURNAL.PONE.0210772.)

- Apiaceae (carrot: https://link.springer.com/chapter/10.1007/978-3-030-03389-7_3 and fennel: https://www.mdpi.com/1422-0067/21/13/4664

- Solanaceae (tomato; https://academic.oup.com/plphys/article/189/2/465/6536974; eggplant: https://www.jstage.jst.go.jp/article/hortj/85/1/85_MI-IR03/_pdf/-char/en) etc etc

Results

106 species name in italics. Double check the entire manuscript

Section 2.1 it is not described within the Methods section. Please describe in an appropriate section the microscopy and the staining methodology.

119 in this section you should also compare the male fertile mtDNA with the mtDNA already sequenced for G. harknessii (NC_027407.1/JX944506.1) and G. Hirsitum (MT: NC_027406.1) Please provide synteny analysis and percentage of conservation. This will help you in understanding if the MF mtDNA you assembled is correct or not.

120 Information on sequencing data are needed both for Illumina and Nanopore. How many millions of reads you got? Which of them were actually mtDNA reads? What was the proportion of cpDNA and nDNA reads? This data would provide important information about the reliability and the efficiency of the mtDNA extraction method.

220 How is possible that an INDEL within the coding region of atp1 did not lead to amino acid sequences change??

266 How did you combine the Illumina RNA-seq and the PacBio RNA-seq? I guess that being two different type of sequencing (short vs long) they provided different results.

266 by having RNA-seq reads (especially the PacBio RNA-seq reads) you should improve the annotation of your mtDNAs. In fact, you could verify whether the genes prediction made previously in silico (section 2.2.) matches with the RNA-seq reads. Please perform this type of analysis.

270 Why did you not mention also the PacBio results?

273 there is not any Table 5 within the manuscript.

274 Please report the number (%) of reads that aligned against the mtDNA so as to have an idea of the percentage of RNA-seq reads that were from the mtRNA.

Discussion

Very poor and mostly a repetition of the results

Methods

Yoy need to cite the companies of each reagent/kit mentioned within this section (company name, city, country)

357 “which were (was? is?) a nuclear restoring line with normal nuclear and normal male fertile of G. harknessii” it is unclear. Please reformulate it. What does “normal nuclear” mean? What does “normal fertile” mean?

359-360 “normal management conditions” is it not acceptable. Please specify all the conditions. Sterility can also depend on environmental conditions (e.g https://www.tandfonline.com/doi/abs/10.1080/15427528.2015.1010680)

361 as far as I understood, buds for RNA-seq analyses were collected from CMS-D2. However, CMS-D2 was not the same CMS plant material used in this study (in line 358 you stated that the CMS line used was SI3A).  Please clarify the entire paragraph 356-362

There is a great confusion between G. harknessii and G. hirsitum throughout the manuscript. Please clarify it because sometimes it is not even clear whether you are working with G.harknessii or G.hirsitum

365 Verb is missing in the first sentence of this paragraph.

365 Method [47] seems to be useful for isolating nuclear DNA from organelles DNA. However, this is not your case, since you were interested in isolating mtDNA from the rest of the DNA). As a matter of fact the method you cited cannot allow the isolation of mtDNA from cpDNA. How did you manage to produce a mtDNA library (as reported in line 369)?

366 qualified how?

370 single end or pair end? Reads length?

375 This is the first time you mention the use of a second sequencing platform (Nanopore). Why is that? You already had the data? (if so, please, provide the repository from which you took the data) You produced also Nanopore data? (if so, add a section to explain how they have been produced, library preparation, etc etc).

377 “using SPAdes v3.110” it is not acceptable. You need to describe all the parameters used. Moreover it is v3.11.0. How Illumina reads were used to correct the sequencing errors made by Nanopore?

378 “using the published homologous species mitochondrial genome”? What does “homologous species” mean? Please clearly mention the species used and all the reference data (repository codes) so that the reader can retrieve the data.

379 This part is unclear. You used blastn to align the newly obtained mitochondrial genome and the mitochondrial genome of an “homologous species”? Why? What is the point of aligning these two mt genomes?

381 “pick out the scaffolds” ? Really? This is not a food recipe, it is supposed to be a scientific paper. Please use the correct form. “Scaffolds were filtered according to […] and those that did not meet the following criteria […] were removed from the dataset.

374-387 This part is totally incomprehensible. Please ask a bioinformatician for help and then rewrite it so that the reader is able to understand the analyses and possibly to repeat them.

388 I suggest you to try the annotation tool GeSeq (Tillich et al., 2017). It is easy to use and provide precise results. In this way, you will have the possibility to compare your annotation pipeline with a different one.

401 This title is grammatically wrong: “compare with”??

411 The title of this paragraph is ambiguous. You did not isolate only mRNA related to mtDNA (since Spectrum kit does not isolate only mtRNA). Therefore it is not clear how you sequenced only the mtRNA.

413 “each were three biological replicates” ??

421 paire reads?

423 clean reads can apply to downstream analsysis?

430 provide all the parameters set for SMRT analysis software.

439 RNA reverse transcription ? Verb is missing.

Author Response

Response to Reviewer 1 Comments

We really appreciate your precious time and insightful comments! Your suggestions are very valuable and helpful for improving our paper. According to your suggestions, we have revised our manuscript carefully. The specific comments and our responses were listed below.

Point 1: Xuan et al, produced a treasure-trove of molecular data by combining second and third generation sequencing analyses. The aim was to assemble the mtDNA of two lines (MF and CMS) of cotton and to combine both genomic and transcriptomic analysis in order to identify possible candidate genes responsible for CMS. Despite the potential value of this work, the manuscript is poorly written, several key-analysis are missing, the Methods section needs to be totally rewritten and some analyses are obscure. There isn’t any major finding nor a real candidate gene is proposed and further investigated. The English form must be deeply edited.

Response 1: Thank you for your comment. We are sorry for the unpleasant reading experience due to our poor English writing. In this revision, we invited a native English-speaking professional in plant science to help polish our article. The manuscript was edited extensively and we hope the revised manuscript could be acceptable to you. In addition, we have modified the text of this manuscript especially the Method and the Discussion section to improve its precision.

Point 2: I'm confident that - after a strong round of revisions - the manuscript will be suitable for a re-consideration. Please ask for assistance for English form and for the bioinformatic section.

Response 2: Thank you for your encourging comments on our manuscript. We hope that our revised manuscript could be considered for accept in this journal.

Point 3: Line 26-29, It is a repetition of what already reported in line 17-19 and 23-24. Please cite within the abstract at least one time the species you are analyzing (G. hirsutum).

Response 3: Thank you for the suggestion. The repeated sentences of lines 25-29 were deleted as recommended and results part was re-organized in the abstract. We are sorry we did not cite the analyzing species (G. hirsutum), we have now added it in lines 17-19.

Point 4: Line 38, This is not exhaustive: CMS is widely applied also in other species. Instead of making a broad description, (that includes monodicots and dicots), I would focus only on eudicots (such us cotton) by providing example of the main families where CMS is widely exploited …

Response 4: We sincerely appreciate the valuable comments. We have checked the literature carefully and added more references on CMS of eudicots into the INTRODUCTION part of the revised manuscript. For example, vegetable brassica, pepper, eggplant, tomato, fennel, carrot and cotton, the corrected reference is in lines 520-537.

Point 5: Line 106 species name in italics. Double check the entire manuscript.

Response 5: Thanks for your careful reading. We have checked the writing formats of the species name in the entire manuscript and made corrections.

Point 6: Section 2.1 it is not described within the Methods section. Please describe in an appropriate section the microscopy and the staining methodology.

Response 6: Thank you for the suggestion, we are sorry we did not provide the methods for section2.1, We have added the information about the microscopy and the staining methodology in the Methods section (lines 394-297.

Point 7: Line 119 in this section you should also compare the male fertile mtDNA with the mtDNA already sequenced for G. harknessii (NC_027407.1/JX944506.1) and G. hiruitum (MT: NC_027406.1) Please provide synteny analysis and percentage of conservation. This will help you in understanding if the MF mtDNA you assembled is correct or not.

Response 7: Thanks for your suggestion. The comparison between the two assembled mtDNA in this study and G. harknessii, G. hirsutum mt genome was conducted and the results were added in the revised manuscript (See line 129-132 and updated Figure 2). Colinearity analysis revealed that the mt genome of 0-613-2R and SI3A was highly homologous with G. hirsutum and G. harknessii, the percentage of conservation was 95.66% and 94.96% in SI3A and 0-613-2R, respectively. The result indicates the two mt-genomes are correctly assembled, and they are well conserved during cross-breeding.

Point 8: Information on sequencing data are needed both for Illumina and Nanopore. How many millions of reads you got? Which of them were actually mtDNA reads? What was the proportion of cpDNA and nDNA reads? This data would provide important information about the reliability and the efficiency of the mtDNA extraction method

Response 8: Thanks for your suggestion. Detailed information about sequence data and mapping rate were added in the manuscript (Lines 120-128). The relevant contents are provided below for your quick reference.

“In the Illumina sequencing, 44.34 and 40.35 million clean reads were respectively obtained from 0-613-2R and SI3A, and the corresponding Q20 values were 97.42% and 97.39%. In the Nanopore sequencing, 2.6 and 2.8 million clean reads were respectively obtained from 0-613-2R and SI3A with total lengths of 14,071 Mb and 14,224 Mb; the lengths obtained for scaffold N50 were 8,199 bp and 7,142 bp. To verify the accuracy of sequencing results, both Illumina and Nanopore sequencing reads were mapped to the cotton G. hirsutum genome (GCA_007990345.1) and the mitochondrial genome (NC_027406.1). Supplementary Table 1 gives the average percentage of cytoplasm DNA (ctDNA) and nuclear DNA (ncDNA) identified in Illumina and Nanopore sequence reads.”

Table S1 the percentage of nuclear and cytoplasmic reads in Illumina and Nanopore sequencing data

Sequencing platform

Short-reads sequencing (Illumina)

Sample

Map to nuclear genome

Map to cytoplasmic genome (in theory)

Map to mitochondria genome

0-613-2R

95.21%

4.79%

2.27%

SI3A

96.42%

3.58%

1.42%

Sequencing platform

Long-reads sequencing (Nanopore)

Sample

Map to nuclear genome

Map to cytoplasmic genome (in theory)

Map to mitochondria genome

0-613-2R

96.08%

3.92%

2.57%

SI3A

94.98%

5.02%

2.70%

*All proportions are calculated using reads with exact alignment ignoring supplementary and secondary alignment. Proportion of reads mapped to cytoplasmic genome is calculated 1 - proportion of reads mapped to nuclear genome.

Point 9: How is possible that an INDEL within the coding region of atp1 did not lead to amino acid sequences change?

Response 9: Sorry for the mistake. The result about the variant within atp1 was rewritten in lines 225-228:

“one insertion of a “T” in the coding region of SI3A atp1 resulted in a frame-shift mutation affecting 44 amino acids (update Figure 4C). Mutations in atp1 may impair ATP synthesis in the mitochondrial respiratory chain. No other InDels resulted in a frame-shift.”

Point 10: How did you combine the Illumina RNA-seq and the PacBio RNA-seq? I guess that being two different type of sequencing (short vs long) they provided different results.

Response 10: Yes, these are two different types of sequencing, the Illumina RNA-seq for short reads sequencing, and the PacBio RNA-seq for long reads sequencing. These two methods were used for different aims. The differential expression analysis of mitochondrial genes was achieved by mapping the Illumina sequencing data to 0-613-2R and SI3A mitochondrial genome, which revealed that nad4 has the lowest expression level in the CMS line SI3A (Figure 6A). Meanwhile, PacBio sequencing reads were mapping to 0-613-2R and SI3A mt genome for discovery of structural variation, for instance nad4 contained intron retention splicing pattern (Figure 6D). So we combined the result of Illumina and PacBio and selected nad4 as a candidate CMS gene in SI3A. This part was shown in the RESULT section 2.5.

Point 11: by having RNA-seq reads (especially the PacBio RNA-seq reads) you should improve the annotation of your mtDNAs. In fact, you could verify whether the genes prediction made previously in silico (section 2.2.) matches with the RNA-seq reads. Please perform this type of analysis.

Response 11: Thanks for your suggestion. The analysis was performed as the follows. The Illumina RNA-seq reads density and Pacbio RNA-seq reads density distribution were compared with the gene density distribution. As shown in the figure below, we can see the majority of the peaks of the three densities were at the same place which indicates that the gene annotations of mtDNA are correctly and properly validated by RNA-seq.

Illumina reads density and Pacbio reads density distribution mapping to the gene density distribution.

Point 12: Why did you not mention also the PacBio results?

Response 12: Thank you very much for the suggestion and we are sorry for not provide detailed information in the original manuscript. Result about PacBio data was added in this revision (Lines 277-279).

Point 13: there is not any Table 5 within the manuscript.

Response 13: Thanks for your suggestion. We are sorry for the mistake and it has been revised (updated table 4, lines 306-307.)

Point 14: Please report the number (%) of reads that aligned against the mtDNA so as to have an idea of the percentage of RNA-seq reads that were from the mtRNA.

Sample

0-613-2R-1

0-613-2R-2

0-613-2R-3

SI3A-1

SI3A-2

SI3A-3

Total raw reads

27,570,398

27,645,300

27,163,449

24,396,971

30,170,978

22,451,704

Total clean reads

26,157,634

26,560,996

26,094,847

23,354,551

28,705,908

21,508,933

Clean reads Q20(%)

97.38

97.40

97.22

97.33

97.13

97.38

Clean reads Q30(%)

92.59

92.66

92.25

92.44

92.08

92.59

Total alignment with mt-genome

25,765 (0.10%)

25,125 (0.09%)

26,713 (0.10%)

16,247 (0.07%)

21,734 (0.08%)

18,759 (0.09%)

Exact alignment with mt-genome

14,708 (0.06%)

14,376 (0.05%)

15,446 (0.06%)

12,592 (0.05%)

16,988 (0.06%)

14,509 (0.07%)

Average (exact)

0.0565%

0.0599%

Response 14: Thanks for your suggestion. The detailed RNA-seq reads alignment results are listed below and provided in our manuscript Table 4. Average alignment reads from Illumina RNA-seq in 0-613-2R and SI3A is 0.56%, and 0.59% (updated Table 4). The small proportiom is due to the fact that we used total RNA for Illumina sequencing and the mitochondrial genome is small, so the alignment reads will be lower than than the reads from nuclear.

Table 4. Mapping results for Illumina RNA-seq data to mitochondrial genome

Point 15:Discussion very poor and mostly a repetition of the results.

Response 15: Thanks for your suggestion. We are sorry for the poor writing. We have made extensive modifications to our entire manuscript, oponions and extra researches on CMS were supplemented in discussion section.

Point 16: You need to cite the companies of each reagent/kit mentioned within this section (company name, city, country).

Response 16: Thank you for pointing this out. We have checked the full manuscript and corrected it.

Point 17: “which were (was? is?) a nuclear restoring line with normal nuclear and normal male fertile of G. harknessii” it is unclear. Please reformulate it. What does “normal nuclear” mean? What does “normal fertile” mean?

Response 17: Thanks for your suggestion. The restorer line 0-613-2R is G. hirsutum with “normal male fertile” means it can be produce fertile pollen, “normal nuclear” mean nuclear genes of 0-613-2R can be restorer the fertility of sterile line SI3A. Sorry for the vague description. We have rewritten the description about G.harknessii in the revised manuscript (Lines 387-389).

Point 18: “normal management conditions” is it not acceptable. Please specify all the conditions. Sterility can also depend on environmental conditions .

Response 18: Thanks for your suggestion. We are sorry for the non-critial description and now it has been revised in the updated manuscript. All the plants were grown in the standard growing conditions with the average tempertaure from 30℃ to 35℃ in Zhejiang province, Hangzhou (Lines 389-391).

Point 19: as far as I understood, buds for RNA-seq analyses were collected from CMS-D2. However, CMS-D2 was not the same CMS plant material used in this study (in line 358 you stated that the CMS line used was SI3A). Please clarify the entire paragraph 356-362

Response 19: Thanks for your critical reading. The cytoplasm source of sterile line SI3A was derived from G. harknessii CMS-D2, so it was named “CMS-D2 line SI3A”. The floral buds for RNA-seq analyses were collected from the CMS-D2 line SI3A and its restorer line 0-613-2R. Sorry for the unclear description about CMS-D2 and SI3A. It has been corrected in the revised manuscript (Lines 392-393).

Point 20: There is a great confusion between G. harknessii and G. hirsitum throughout the manuscript. Please clarify it because sometimes it is not even clear whether you are working with G.harknessii or G.hirsitum.

Response 20: Thanks for your suggestion. Sorry for this vague description. The plant materials used in this study were the G. hirsutum restoring line 0-613-2R and an upland CMS-D2 line SI3A with cytoplasm from G. harknessii. It has been corrected in the revised manuscript (Lines 387-388).

Point 21: Line 365 Verb is missing in the first sentence of this paragraph.

Response 21: Thanks for your careful reading. We are sorry for this grammatical error and it was corrected in the revised manuscript.

Point 22: Method [47] seems to be useful for isolating nuclear DNA from organelles DNA. However, this is not your case, since you were interested in isolating mtDNA from the rest of the DNA). As a matter of fact the method you cited cannot allow the isolation of mtDNA from cpDNA. How did you manage to produce a mtDNA library (as reported in line 369)?

Response 22: Thanks for your suggestion. we are sorry for making a mistake about the DNA extraction reference. In fact we extracted the total genomic DNA for Illumina sequencing and Nanopore sequencing by a modified CTAB method. The method reference has been revised in the updated manuscript.

Point 23: Line 366: qualified how?

Response 23: The purity: (1.8<OD260/280<2.0) (2.0<OD260/230<2.2), DNA integrity was examined by agarose gel electrophoresis, with clear primary bands and no apparent degradation.

Point 24: Line 370: single end or pair end? Reads length?

Response 24: Sequencing was performed on the Illumina Novaseq 6000 at the Paired-end model with 2×150 bp reads length. Detailed information about library construction and sequencing for Illumina, PacBio and Nanopore sequencing were added in Methods section in the lines 402-407.

Point 25: This is the first time you mention the use of a second sequencing platform (Nanopore). Why is that? You already had the data? (if so, please, provide the repository from which you took the data) You produced also Nanopore data? (if so, add a section to explain how they have been produced, library preparation, etc etc).

Response 25: We produced Illumina sequencing and Nanopore sequencing to assemble the mt-genomes to ensure the accuracy of the assembly. All sequencing data will released after the manuscript is accepted. The library preparation and Nanopore sequencing processing were provided in the updated manuscript in the lines 401-410.

Point 26: “using SPAdes v3.110” it is not acceptable. You need to describe all the parameters used. Moreover it is v3.11.0. How Illumina reads were used to correct the sequencing errors made by Nanopore?

Response 26: SPAdes v3.11.0 was used to assemble the mt genome using the default parameters. After the raw reads filter, the Illumina sequencing clean data were de novo assembled by SPAdes v3.11.0, Nanopore sequencing data were aligned back to the Illumina assembled contigs, removed some incorrect locus by Nanopore sequencing reads, then segregated the aligned reads and reassemble them.

Point 27: “using the published homologous species mitochondrial genome”? What does “homologous species” mean? Please clearly mention the species used and all the reference data (repository codes) so that the reader can retrieve the data.

Response 27: We are sorry that we didn’t provide the details about the homologous mt genome. We used the homologous species Gossypium arboreum (NCBI: KR736342) mt genome for filtering the scaffolds. Specific information has been modified in line 419.

Point 28: This part is unclear. You used blastn to align the newly obtained mitochondrial genome and the mitochondrial genome of an “homologous species”? Why? What is the point of aligning these two mt genomes?

Response 28: Thanks for your comments. In this study, we used total DNA for Illumina and Nanopore sequencing for assembly. The Illumina sequencing data were de novo assembled then aligned to the homologous mitochondrial genome (Gossypium arboreum), selected the contigs with homology to G. arboreum mt genome as the mitochondrial contigs for the next extension and assembly, fragments that are of low coverage and obviously not the target genome were removed. Briefly, the mitochondrial genome contigs were filtered from the assembled Illumina contigs by aligned with the homologous mitochondrial genome.

Point 29: “pick out the scaffolds” ? Really? This is not a food recipe, it is supposed to be a scientific paper. Please use the correct form. “Scaffolds were filtered according to […] and those that did not meet the following criteria […] were removed from the dataset.

Response 29: Thanks for your suggestion. We are sorry for using the informal description, it has been revised in the updated manuscript. “the scaffolds were filtered according to the results of BlastN and Exnerate, sequences with e-value higher than 1e-10 and protein identity lower than 70% were removed from the dataset.”

Point 30: This part is totally incomprehensible. Please ask a bioinformatician for help and then rewrite it so that the reader is able to understand the analyses and possibly to repeat them.

Response 30: Thanks for your suggestion. Sorry for the poor description. This part has been rewritten and checked by the bioinformatician.

Point 31: I suggest you to try the annotation tool GeSeq (Tillich et al., 2017). It is easy to use and provide precise results. In this way, you will have the possibility to compare your annotation pipeline with a different one.

Response 31: Thanks for your suggestion. The principle of the annotation software Motify we use is consistent with GeSeq you provided, which is to search for genes by homology with the reference sequence and perform the subsequent manual correction. So by theory the GeSeq annotation should be very similar to Motify. For examination we made a draft annotations for 0-613-2R and SI3A mt-genomes using Geseq, we found that the results of the two tools were mostly consistent (as shown in Figures below). For those inconsistent gene features, they were mostly non-coding RNA fragments with extremely short length, which would be removed by the following manual check and QC.

Mitochondrial genome map of the restorer line 0-613-2R

Mitochondrial genome map of the CMS line SI3A-mt1

Mitochondrial genome map of the CMS line SI3A-mt2

Figure 3. Mitochondrial genome map of the CMS line SI3A-mt3

Point 32: Line 401: This title is grammatically wrong: “compare with”??

Response 32: Thank you for pointing out this error, it has been corrected. “Differential analysis between SI3A and 0-613-2R mitochondrial genome”

Point 33: The title of this paragraph is ambiguous. You did not isolate only mRNA related to mtDNA (since Spectrum kit does not isolate only mtRNA). Therefore it is not clear how you sequenced only the mtRNA.

Response 33: Sorry for the ambiguous description. ”mitochonrial transcriptome sequencing” is an error description. Actually we isolated the total RNA and subjected for Illumina and full-length PacBio transcriptome sequencing.

Point 34: “each were three biological replicates” ??

Response 34: Sorry for the ambiguous description. It means the total RNA from the floral buds (4-5mm size, abortion stage) of SI3A and 0-613-2R in each with three biological repeats for the subsequent sequencing.

Point 35: paire reads?

Response 35: we are sorry for the spelling mistake, it’s the “paired-end” and corrected in the updated manuscript.

Point 36: clean reads can apply to downstream analsysis?

Response 36: Thanks for your suggestion, we are sorry for the informal description.

The original Illumina reads are processed by data quality control to generate the clean reads for the gene expression qualification.

Point 37: Provide all the parameters set for SMRT analysis software.

Response 37: All the parameters set for SMRT analysis software were added. “parameters: min length 300, min accuracy 0.75, min passes 0.”

Point 38: RNA reverse transcription ? Verb is missing.

Response 38: Sorry for the mistake, it has been corrected.

Finally, special thanks to you for your precious time and your constructive comments!

Reviewer 2 Report

The manuscript title “Comparison of mitochondrial genomes between a cytoplasmic male-sterile line and its restorer line in G. hirsutum for identifying candidate CMS genes” described the comparative analysis of the mitochondrial genomes of two cotton lines (CMS line and restorer line). The genomic analysis is supplemented with RNA-seq data to illustrate the CMS and Rf gene interaction. The evidence provided are not strong enough to declare the I gene nad4 as a casual gene of sterility. However, the authors generated well-enough data for future studies. The manuscript is written well. I recommend the manuscript for the acceptance in IJMS after the minor revisions.

Comments;

Check the space between G. hirsutum in the title. The same is need to be corrected throughout the text.

Italicize the scientific and gene names in the text. One example is line 106 and 139.

Please explain a little more how the de novo hybrid genome assembly was done?
